# The Journey of the Default Mode Network: Development, Function, and Impact on Mental Health

**DOI:** 10.3390/biology14040395

**Published:** 2025-04-10

**Authors:** Felipe Rici Azarias, Gustavo Henrique Doná Rodrigues Almeida, Luana Félix de Melo, Rose Eli Grassi Rici, Durvanei Augusto Maria

**Affiliations:** 1Graduate Program in Medical Sciences, School of Medicine, University of São Paulo, São Paulo 05508-220, SP, Brazil; felipe.azarias@outlook.com; 2Graduate Program in Anatomy of Domestic and Wild Animals, College of Veterinary Medicine and Animal Science, University of São Paulo, São Paulo 05508-220, SP, Brazil; gustavohdra@usp.br (G.H.D.R.A.); biologafelix@gmail.com (L.F.d.M.); roseeligrassirici@gmail.com (R.E.G.R.); 3Graduate Program in Structural and Functional Interactions in Rehabilitation, School of Medicine, University of Marília (UNIMAR), Marília 17525-902, SP, Brazil; 4Development and Innovation Laboratory, Butantan Institute, São Paulo 05585-000, SP, Brazil

**Keywords:** default mode network, neuropsychology, brain connections

## Abstract

The Default Mode Network (DMN) is a brain network that becomes active when the brain is at rest. It is crucial for processes like self-reflection, emotional processing, social interaction, and mental exploration. Research has shown that the DMN is not exclusive to humans but is also found in non-human animals, suggesting its evolutionary importance across species. The DMN plays a significant role in brain development, especially during childhood and adolescence, where it shapes cognitive and emotional abilities. This review explores the structural, functional, and evolutionary aspects of the DMN and its impact from early development through adulthood. Understanding how the DMN operates can deepen our knowledge of how the brain supports thinking, behavior, and mental health. This knowledge can, in turn, lead to more effective treatments for various neuropsychiatric conditions. By studying the DMN, we gain insights into the neural mechanisms behind cognition and behavior, which could help develop better therapeutic strategies for mental health disorders.

## 1. Introduction

It was formerly believed that the brain remained inactive during rest. Resting data were primarily collected for comparison with other cognitive tasks [1]. However, subsequent research challenged this notion. A meta-analysis of brain regions revealed increased activity in certain areas during passive periods [2]. This led to the discovery of the DMN through functional neuroimaging studies, particularly using techniques like functional magnetic resonance imaging (fMRI). These studies allowed researchers to map synchronized and coherent brain activity during rest or when not engaged in specific tasks [1,3,4,5].

Since its identification by Shulman and colleagues at the end of the 20th century, the DMN has become a focal point across disciplines—from neuroscience to psychology and medicine. Initially, the DMN piqued curiosity due to its activity during rest, leading to investigations into its role in internal cognitive processes and states of consciousness [6]. Advances in neuroimaging technology, particularly fMRI, have allowed for a deeper understanding of DMN activity and connectivity, revealing its involvement in various brain functions [6,7].

Notably, beyond its role in introspection and self-reflection, the DMN plays a crucial part in episodic memory, emotional processing, and the construction of mental narratives [5,8,9]. However, the DMN does not operate in isolation but interacts dynamically with other large-scale brain networks, enabling the brain to adapt to varying emotional and cognitive demands. For instance, the executive control network (ECN), responsible for goal-directed behavior, cognitive control, and decision-making, exhibits a reciprocal relationship with the DMN: during high cognitive demand, DMN activity decreases while ECN activity increases, facilitating the transition between introspective and task-focused states [10].

Additionally, the Salience Network (SN) plays a crucial role in detecting relevant stimuli and processing emotions, acting as a switch between the DMN and ECN. When a salient stimulus is identified, the SN deactivates the DMN and activates the ECN, ensuring that the brain prioritizes essential information and effectively regulates affective states [11].

Recent studies have increasingly emphasized the dynamic nature of brain network connectivity, particularly within the DMN. Far from being static, the connectivity of the DMN varies over time in response to fluctuations in emotional and cognitive states. A study by Lettieri et al. [12] demonstrates how large-scale brain networks, including the DMN, exhibit connectivity patterns that are not fixed but instead align with ongoing affective states. The research highlights that these fluctuations occur at multiple timescales, suggesting that the DMN’s connectivity is intricately linked to both emotional and cognitive demands. Specifically, the study found that the DMN, which is typically associated with self-referential thought and rest, shows dynamic shifts in its connectivity in response to affective states. This finding challenges the traditional view of the DMN as a network of fixed, stable connectivity and underscores its role in adapting to changes in mental states, such as emotional fluctuations or shifts in cognitive processing. These findings offer valuable insights into the flexible and responsive nature of brain network connectivity, particularly in relation to the DMN, reinforcing the idea that brain activity is shaped by both internal and external demands.

During resting periods, the DMN exhibits strong internal connectivity, whereas emotional or cognitive demands trigger dynamic changes that activate the ECN and SN [13]. These fluctuations allow the brain to track and respond to affective states across different timescales. In the short term, rapid shifts in connectivity reflect immediate responses to emotional stimuli, such as sudden threats or unexpected positive events [11]. In the long term, connectivity patterns adapt to sustained emotional states, such as chronic stress or prolonged well-being [13]. Thus, the dynamic interaction among the DMN, ECN, and SN is essential for cognitive and emotional flexibility, contributing to mental health maintenance and adaptation to environmental challenges [5,8,9].

Recent studies have explored the DMN’s links to neuropsychiatric disorders such as depression, anxiety, and attention deficit hyperactivity disorder (ADHD), offering new diagnostic and treatment perspectives [6,14,15]. Overall, the multidisciplinary research on DMN underscores its potential to revolutionize our understanding of brain function and its impact on mental health and well-being (Figure 1).

While a comprehensive understanding of the evolution of the DMN remains an ongoing area of research, several theories have emerged to elucidate its purpose. Below, we outline the 10 primary theories regarding the DMN’s function. Notably, the two prevailing hypotheses are the sentinel hypothesis [3,17] and the internal mental activity hypothesis [1,4,5,49].

The sentinel hypothesis suggests that, even at rest, our brain actively monitors the environment, processing information from our surroundings and preparing us to react to relevant stimuli, which is crucial for our survival and adaptation. This means that even when we are not actively engaged in external tasks, the DMN remains active, processing information from the environment around us and preparing us to respond to relevant stimuli. This continuous activity of the DMN can be vital in terms of survival and adaptation, allowing us to react quickly to possible threats or opportunities [17].

The internal mental activity hypothesis, also known as spontaneous cognition, is a theory that describes the role of the DMN in the generation and manipulation of internal thoughts during periods of rest and introspection. This hypothesis was first proposed by Christoff et al. [49]. According to this theory, when individuals are not engaged in external or externally directed tasks, the DMN becomes active, facilitating reflection on the self, the retrieval of personal memories, the mental simulation of past and future events, as well as the planning of imaginary scenarios. Internal mental activity involves processes such as self-awareness, self-reflection, and mental prospection, in which individuals explore and elaborate on their own thoughts, feelings, and experiences. This hypothesis highlights the importance of DMN in facilitating spontaneous cognition and building a more elaborate self-awareness, contributing to the understanding of internal mental processes and self-awareness [49].

The Associative Memory Theory, developed by Brian Levine and Endel Tulving in pioneering studies in the field of neuropsychology, proposes that human memory is organized in networks of associations between elements of information. According to this theory, events are coded and stored not in isolation but in conjunction with other related elements, thus forming a complex network of interconnected memories. These associations can be activated by contextual stimuli or cues, facilitating the retrieval of associated information. Associative Memory Theory has been fundamental to our understanding of how the human brain processes, stores, and retrieves information and has significant implications for various fields, from neuroscience to education [50].

The self-construction hypothesis suggests that the DMN is involved in the continuous and dynamic construction of one’s sense of identity and self-concept. According to this theory, the DMN is actively involved in the construction of personal narratives, in which past experiences are constantly re-evaluated and re-interpreted, thus contributing to the formation of the individual’s identity and self-concept. Dennett argues that the DMN plays a fundamental role in creating a coherent and continuous narrative of our lives, helping us to understand who we are and how we relate to the world around us. This hypothesis emphasizes the importance of the DMN in self-reflection and in the active construction of consciousness and personal identity [51].

The social connection hypothesis proposes that the DMN is intrinsically linked to processing social information and understanding other people’s minds. This theory postulates that DMN activity is involved in the interpretation of intentions, empathy, and the theory of mind, which are processes that are essential for social interaction. Functional neuroimaging studies have shown a significant overlap between DMN regions and areas activated during tasks involving social interaction, such as observing facial expressions and attributing mental states to other individuals. In addition, clinical research with patients with autism spectrum disorders, who often have difficulties in social interactions, has shown alterations in the connectivity and activity of the DMN. This evidence supports the idea that the DMN plays a fundamental role in perceiving and understanding social interactions, providing a neurobiological basis for social cognition [19].

Autobiographical Memory Hypothesis suggests that the DMN plays a key role in the retrieval and consolidation of autobiographical memory. This theory posits that the DMN is particularly involved in the retrieval of personal memories and past experiences, allowing individuals to reflect on autobiographical events and construct coherent narratives about the self over time. According to this hypothesis, the DMN acts as a neural hub for the reactivation of autobiographical memories during rest and introspection, facilitating reflection on the past and the construction of personal identity. In addition, autobiographical memory plays a crucial role in self-reflection, self-awareness, and the construction of a continuous life narrative. Thus, the Autobiographical Memory Hypothesis highlights the importance of the DMN in integrating past experiences in the construction of personal identity and in maintaining the continuity of the self over time [52].

The creative imagination hypothesis suggests that the DMN is involved in facilitating creativity and generating original ideas. According to this hypothesis, when a person is at rest or involved in tasks that do not require external attention, the DMN becomes active, allowing the mind to wander and explore different mental scenarios. This not only aids reflection on past experiences but also the construction of mental narratives about the future. DMN activity during these periods of creative imagination can promote the association of seemingly disconnected concepts, leading to insights and innovative solutions. This hypothesis suggests that the DMN plays a fundamental role in the human creative process, providing a neural substrate for generating new ideas and solving complex problems [19].

The self-reflection hypothesis suggests that the DMN is involved in self-reflection and introspection. When we are not focused on external tasks, the DMN facilitates reflection on ourselves and our past and future experiences, allowing us to imagine and plan possible scenarios. This self-reflective activity is essential for the development of sophisticated self-awareness and the ability to mentally simulate different situations [19].

The Mental Foresight Hypothesis is a theory that postulates that the DMN plays a crucial role in simulating future events and drawing up hypothetical scenarios in the human mind. According to Conway, the ability to mentally envision future events is essential for planning, adaptation, and decision-making. By facilitating mental prospecting, DMN allows individuals to anticipate and prepare for situations that have not yet occurred, contributing to better adaptation to the environment and the development of effective coping strategies. In addition, mental prospecting is intrinsically linked to episodic memory, allowing individuals to recover and recombine elements of past experiences in order to construct plausible future scenarios. Therefore, the Mental Foresight Hypothesis highlights the importance of the DMN not only in reflecting on the past but also in preparing for the future, providing a neural basis for building expectations and making plans for events yet to come [53].

The temporal coherence hypothesis, also known as the temporal synchronization theory, is a perspective that suggests that the DMN plays a crucial role in integrating and organizing mental events over time. According to this theory, the DMN functions as a neural network that facilitates the connection between different temporal moments of human experience, allowing for the fluid transition between past, present, and future events in an individual’s mind. DMN activity during periods of rest and introspection helps in the consolidation of episodic memory, the retrieval of stored information, and the mental projection of future events based on past experiences. In addition, the DMN plays a fundamental role in constructing cohesive mental narratives and maintaining a sense of continuity and personal identity over time. This hypothesis highlights the importance of the DMN in the temporal organization of human cognition and in the creation of a unified narrative of individual experience [25].

While there exist multiple theories regarding the function of the DMN, some of these theories share common characteristics. Similarly, various research avenues explore the DMN (Table 1).

## 2. Neuroanatomical Basis of the Default Mode Network

The Default Mode Network is a large-scale brain network comprising key regions such as the anterior medial frontal cortex, ventral medial prefrontal cortex, posterior cingulate cortex, precuneus, inferior parietal lobule, and middle temporal gyrus [25]. These areas exhibit heightened neuronal activity during wakeful rest and mind-wandering, playing a fundamental role in self-referential thought, autobiographical memory, and social cognition [8,63].

Although the DMN is typically deactivated during externally directed tasks, research suggests that its activity persists or re-emerges intermittently, often competing with task-specific processing. This dynamic interplay has been linked to attentional fluctuations, performance deficits, and cognitive lapses during goal-directed tasks [64]. Additionally, the DMN is not a monolithic entity but consists of functionally specialized subsystems.

The medial prefrontal cortex (mPFC) is primarily implicated in self-referential processing and affective decision-making, whereas the posterior cingulate cortex (PCC) and precuneus contribute to memory retrieval and the integration of sensory and cognitive information [65]. Additionally, interactions between the DMN and other large-scale networks, such as the Salience Network (SN) and the Central Executive Network (CEN), play a crucial role in cognitive and behavioral regulation [66,67]. Disruptions in these interactions have been linked to neuropsychiatric conditions, including major depressive disorder, schizophrenia, and Alzheimer’s disease [68]. Functionally, the DMN can be subdivided into three principal subsystems: (1) the central core, (2) the medial temporal subsystem, and (3) the medial prefrontal subsystem [8]. Each subsystem contributes uniquely to cognitive function and behavioral regulation.

These regions are critical for introspection, autobiographical memory retrieval, and self-referential thought:Posterior Cingulate Cortex (PCC): The PCC functions as a highly connected node within the DMN, facilitating the communication between its subsystems. It is implicated in self-reflective processing, the monitoring of internal and external stimuli, and the integration of autobiographical information [69]. Aberrant activity within the PCC, particularly hyperactivity, has been associated with psychiatric conditions such as depression and excessive rumination [69];Precuneus: Situated in the medial parietal lobe, the precuneus is involved in the consolidation of episodic memory, imaginative cognition, and the simulation of future events [70]. Furthermore, it plays a role in attentional shifts and transitions between wakefulness and mind-wandering states.

The medial temporal subsystem of the DMN is primarily engaged in episodic memory retrieval and the mental construction of prospective scenarios. The key components of this subsystem include:Hippocampus and Parahippocampal Cortex: Although the hippocampus is not typically considered part of the DMN, it is closely linked to it. Together, they play vital roles in human cognition, impacting memory, spatial navigation, introspection, and other cognitive processes [25,71]. The interaction between these two neuroanatomical structures has garnered increasing interest in neuroscience, with significant implications for understanding neurological disorders and developing therapeutic interventions [25,71].

These structures are integral to autobiographical memory retrieval and future event simulation. The hippocampus interacts with the PCC and precuneus to integrate past experiences into coherent, prospective simulations [72].

Recent neuroscientific research has uncovered a dynamic interaction between the DMN and the hippocampus across various cognitive conditions and mental states. For instance, during periods of rest or mind-wandering, DMN activity tends to rise, indicating its involvement in introspection and mental projection into the future [25]. Concurrently, the hippocampus is crucial for the formation and retrieval of memories, including autobiographical and spatial memories [73,74,75].

Functional neuroimaging research has revealed a significant overlap between brain regions activated during memory tasks and those associated with the DMN, including the hippocampus [72]. This suggests that the DMN may be involved in the retrieval and consolidation of autobiographical memories, while the hippocampus is key for encoding and retrieving specific memories.

Moreover, studies in both animal models and humans have shown that the hippocampus is crucial for spatial navigation and the mental representation of environments, which are closely linked to the DMN’s role in simulating future scenarios [74,76]. Understanding the interaction between the DMN and the hippocampus has important implications for research with children and for the diagnosis and treatment of neurological disorders. In developing children, early identification of alterations in connectivity between the DMN and hippocampus can provide insights into normal cognitive development and identify potential early markers of neurological disorders such as autism spectrum disorder (ASD) and attention deficit hyperactivity disorder (ADHD) [75,77].

Furthermore, in neurological conditions such as temporal lobe epilepsy, which directly affects the hippocampus, understanding the interaction between the DMN and the hippocampus may help elucidate the mechanisms underlying the alterations in cognition and memory observed in these patients. These insights may inform the development of new therapeutic approaches aimed at modulating the activity of the DMN and hippocampus to improve the cognitive symptoms associated with these conditions.

Amygdala: The amygdala is a fundamental region for processing emotions and emotional memories. Situated bilaterally in the medial temporal lobes, next to the hippocampus, it consists of several sub-regions, including the basolateral nucleus, central nucleus, medial cortical nucleus, and lateral nucleus [11,75,78].

Research has highlighted the amygdala’s functional complexity. Beyond its primary role in evaluating and reacting to emotional stimuli, it also regulates the autonomic and endocrine nervous systems in response to these stimuli [79]. Functional neuroimaging studies have shown that different sub-regions of the amygdala may specialize in processing various emotions, such as fear, anger, and pleasure [80].

The amygdala’s connections with other brain regions are vital for its function. It has extensive links with the prefrontal cortex, cingulate cortex, hippocampus, and thalamus, enabling it to integrate sensory and emotional information to evaluate the emotional significance of stimuli and generate appropriate behavioral responses [81]. Functional connectivity studies have also highlighted the importance of connections between the amygdala and the medial prefrontal cortex in regulating emotional responses [11,75,82].

Recent neuroscientific research has underscored the amygdala’s pivotal role in emotional regulation and its connection with the DMN, a neural network integral to self-reflection, memory, and future scenario projection. The amygdala, a subcortical structure located bilaterally in the medial temporal lobe, is crucial for processing emotional stimuli and generating emotional responses [11,75].

Functional neuroimaging studies, including fMRI, have demonstrated the dynamic interplay between the amygdala and the DMN during emotional processing. Research indicates that heightened amygdala activity correlates with decreased DMN activity, suggesting an inverse relationship between these networks [77,83,84,85].

Moreover, studies on patients with psychiatric disorders such as anxiety, depression, and post-traumatic stress disorder (PTSD) have shed light on alterations in amygdala function and connectivity. For instance, individuals with these disorders often exhibit amygdala hypersensitivity and DMN dysfunction, contributing to heightened emotional symptoms [86].

Other studies, such as the one carried out by Ghashghaei and Barbas [87], provided evidence that the amygdala maintains dense, bidirectional connections with various cortical regions, including the dorsal medial prefrontal cortex (DMPC) [87]. These connections are essential for emotional regulation and the integration of emotional information with higher cognitive processes. In addition, the amygdala is also involved in modulating attention and evaluating environmental stimuli in terms of their emotional relevance [80,88]. This capacity for emotional evaluation and attribution of meaning is fundamental to behavioral adaptation and survival. In summary, the amygdala plays a crucial role in emotional regulation, interacting dynamically with the DMN and other neural networks. Its dysfunction is associated with a variety of psychiatric disorders, highlighting the importance of understanding its anatomy, function, and connections in normal and pathological contexts [88,89].

Orbitofrontal Cortex: This region is involved in reward evaluation and emotion-based decision-making. Its extensive connections with the limbic system allow for the regulation of emotional responses and contextual representations [90].

The medial prefrontal subsystem contributes to affective processing, self-referential cognition, and the regulation of emotional responses. It consists of the following:Medial Prefrontal Cortex (mPFC): The mPFC is central to self-referential processing and emotional regulation, integrating social and affective information [91]. Dysfunctions in this region have been implicated in psychiatric disorders, including depression and anxiety;Orbitofrontal Cortex: This region is involved in reward evaluation and emotion-based decision-making. Its extensive connections with the limbic system allow for the regulation of emotional responses and motivational states, influencing decision-making processes [90].

The DMN plays a crucial role in self-referential cognition, memory processing, and emotional regulation. Its dynamic interactions with other neural networks underpin key cognitive functions, and its dysfunction is implicated in various neuropsychiatric conditions. Future research should further explore the causal mechanisms linking DMN alterations to psychopathology, as well as potential therapeutic interventions targeting these neural circuits. Understanding the intrinsic organization of the DMN and its broader network interactions is essential for advancing the neuroscience of cognition and mental health (Figure 2).

## 3. Default Mode Network and Brain Networks

The Default Mode Network does not operate in isolation; it is interconnected with other brain networks and shows dynamic activity patterns at rest and during specific tasks. Some important connections include the following.

### 3.1. Attention Networks

Attention networks, such as the executive control network, exhibit an anti-correlation with the DMN [92] and play a role in the internal orienting of attention, while attention networks are involved in external orienting [20].

Additional studies have highlighted the importance of executive control in modulating DMN activity during specific cognitive tasks [55]. Interactions between the attention network and the DMN represent a crucial aspect of the brain’s functional organization, directly influencing complex cognitive and behavioral processes. Although historically considered antagonistic networks, recent studies have revealed a dynamic interdependence between these systems, essential for cognitive flexibility and behavioral adaptation [6,71,93].

When performing cognitively challenging tasks, such as problem-solving or decision-making, there is a selective deactivation of the DMN by the attention network [94]. This deactivation is crucial for the efficient allocation of neural resources to high-level cognitive processes, allowing greater focus and directing attention to relevant stimuli in the environment [8]. For example, functional neuroimaging studies have shown that, in healthy individuals, deactivation of the DMN while performing demanding tasks is associated with more effective cognitive performance and greater problem-solving ability [95].

However, the interactions between the attention network and the DMN are influenced by various contextual factors. For instance, during acute or chronic stress, the ability to deactivate the DMN may be compromised [66,96]. Chronic stress can lead to prolonged DMN activation, even during cognitively demanding tasks, resulting in concentration difficulties and attention lapses. Additionally, inadequate sleep or the presence of psychiatric disorders, such as anxiety and depression, can negatively impact the interactions between the attention network and the DMN [66,96]. In summary, understanding these interactions is essential for attention regulation, cognitive processing, and behavioral adaptation. Comprehending these interactions and the influencing factors is essential for designing therapeutic interventions targeting neuropsychiatric disorders and enhancing cognitive performance across diverse contexts.

### 3.2. Limbic System

The limbic system plays a crucial role in regulating emotions and motivation. Understanding this connection is essential for unraveling the mechanisms underlying emotional regulation and the stress response. Within the limbic system, several interconnected brain structures, including the amygdala, hippocampus, hypothalamus, and anterior cingulate cortex, contribute to these processes.

The amygdala, in particular, serves as a central player in emotional processing and the formation of memories associated with emotions. Research indicates that the amygdala evaluates and responds to emotionally charged stimuli, triggering appropriate physiological and behavioral reactions in situations involving danger or reward [81].

Additionally, the hippocampus, another key component of the limbic system, is responsible for memory formation and retrieval, especially related to emotionally significant events. Its interaction with the DMN influences the encoding and retrieval of autobiographical memories, contributing to the development of a cohesive self-narrative over time [79,96].

The interconnections between the DMN and the limbic system serve as a neurobiological foundation for emotional regulation and stress response. Understanding how these networks interact and mutually influence each other is essential for comprehending various psychological phenomena, including anxiety disorders, depression, decision-making processes, and emotional regulation [88,97,98].

The complex interplay between the DMN and the limbic system highlights the integrated and dynamic nature of human brain function, offering valuable insights into the intricacies of the human mind and behavior.

### 3.3. Salience Network

The Salience Mode Network (SN) and the Default Mode Network are two brain systems that play key roles in regulating attention, emotional processing, and decision-making. The SN is responsible for detecting and directing neural resources toward salient stimuli in the environment, while the DMN is associated with introspection, autobiographical memory, and internal processing [71,99,100,101].

The interaction between the SN and the DMN is crucial for cognitive and behavioral flexibility, allowing the brain to switch between states of activation and deactivation according to the demands of the environment [67]. During situations of rest or introspection, the DMN tends to be more active, while the SN exhibits less activity. On the other hand, in response to salient stimuli or cognitive challenges, the SN is activated to direct attention and suppress DMN activity [67].

Functional neuroimaging studies have demonstrated an overlap between SN and DMN regions, particularly in the anterior cingulate cortex and anterior insular cortex [99]. These regions play a central role in integrating salient signals from the environment and regulating attention and alertness.

In addition, the interconnectivity between the SN and the DMN is mediated by several other brain regions, including the dorsolateral prefrontal cortex and the inferior parietal cortex [77]. These connections enable efficient communication between the two systems and facilitate the coordination of neural activities in response to external and internal stimuli.

Understanding the mechanisms underlying the interaction between the SN and the DMN is crucial for the development of therapeutic interventions aimed at neuropsychiatric disorders in which these systems are dysfunctional, such as attention deficit hyperactivity disorder (ADHD), autism spectrum disorder (ASD), and anxiety disorders [102].

In summary, the dynamic interaction between the Salience Network and the Default Mode Network plays a key role in the regulation of attention, emotional processing, and behavioral adaptation. Understanding these systems and their interaction can provide valuable insights for the development of more effective treatments for a variety of neuropsychiatric disorders.

## 4. Default Mode Network Investigation Tools

The study of the brain in a resting state must take place in the absence of stimuli from the outside world, in an unintentional state that is often unnoticed and difficult to monitor, replicate, or report [103]. Studies have shown that certain brain areas are activated synchronously in certain mental processes, and to describe this process, the term functional connectivity (FC) was created, with the set of synchronous areas being called a network [103]. A network can also be defined as a set of nodes or vertices and the connections (or graphs) established between these vertices [104]. In fMRI studies, the nodes are typically sets of voxels, and the connection between them is the correlation of the BOLD signal between the time series of each node [104]. The common characteristic of brain networks is that they reflect a small-world architecture, i.e., the composition of nodes and their connections in the same network must be stronger than the connection with nodes belonging to other networks. This implies a greater speed of information transmission between the nodes [105]. The study of brain connectivity has been widely explored in healthy individuals and can provide information about human behavior and how this organization is altered for certain individual differences [103].

The investigation of DMN has advanced significantly with the development of various neuroimaging techniques. Each method provides unique insights into the functional activity and connectivity of this brain network, contributing to a broader understanding of its structure and function.

Functional magnetic resonance imaging offers high spatial resolution, allowing precise identification of brain regions active during specific tasks or at rest. This technique is fundamental for mapping brain activity and understanding the interactions between different DMN regions. Despite its high spatial resolution, fMRI has limited temporal resolution, making it difficult to capture rapid neuronal activity dynamics. Additionally, it is sensitive to motion artifacts and depends on the hemodynamic response, which may not directly reflect neuronal electrical activity.

Acharya et al. [106] used fMRI to identify affected regions within the DMN in individuals with Mild Cognitive Impairment (MCI). The authors applied a novel node significance score to quantify the disparity between healthy and MCI groups, highlighting significant alterations in the posterior cingulate cortex and fusiform gyrus.

Electroencephalography (EEG) and Magnetoencephalography (MEG) offer alternatives to fMRI by providing direct measurements of neuronal activity with high temporal resolution [107]. EEG, through the analysis of brain oscillations and correlations with the DMN, has been used to investigate resting states and transitions between different cognitive states [107,108]. MEG, in turn, provides better spatial accuracy than EEG, allowing for a more refined mapping of DMN functional activity.

However, these techniques also present challenges. The low spatial resolution of EEG limits the precise identification of brain regions involved in the DMN, while MEG is less sensitive to neural activity in deep structures, such as the hippocampus, which is essential for memory functions related to the DMN [109]. Additionally, both techniques are highly susceptible to physiological and environmental artifacts, requiring the use of sophisticated data filtering techniques [63].

Using MEG, a study in 2022 analyzed DMN functional connectivity in patients with fibromyalgia using MEG. The results indicated significant alterations in the activity of the primary somatosensory cortex, suggesting a possible contribution to widespread pain perception in these patients. In 2021, a study used EEG to assess language development in preterm-born children. The study revealed alterations in DMN functional connectivity associated with linguistic deficits, emphasizing the importance of early monitoring in at-risk populations.

The Positron Emission Tomography (PET) provides a complementary perspective on the study of the DMN by enabling the investigation of brain metabolism and the neurochemical mechanisms underlying its activity [71,110]. This approach has been valuable in exploring DMN alterations associated with neuropsychiatric disorders such as Alzheimer’s disease and depression [94,111]. However, PET is an invasive technique, as it requires the administration of radiopharmaceuticals, which limits its applicability for longitudinal studies and in healthy populations. Additionally, its temporal resolution is low, making the analysis of dynamic processes more challenging compared to fMRI and electrophysiological techniques [112].

Some of the main tools used to investigate the DMN are summarized in Table 2, highlighting their strengths and limitations.

The integration of multiple neuroimaging techniques is essential to overcome the individual limitations of each method and provide a more comprehensive understanding of the DMN [113]. Recent advances in combining fMRI with EEG/MEG have allowed for better correlation between electrical activity and hemodynamic responses, offering more detailed insights into the temporal and spatial dynamics of the DMN [114].

When combined in integrated studies, these techniques provide a comprehensive understanding of the activity and connectivity of the DMN and how it relates to different mental states, neuropsychiatric disorders, and therapeutic interventions [115]. Furthermore, emerging techniques such as ultra-high-resolution fMRI promise to enhance the detection of functional and structural patterns of the DMN with greater precision [63].

Thus, several studies have reported evaluating the influence of various variables on connectivity, such as gender [116], age [54,116,117], intelligence [118,119,120], psychoactive intake [121], and meditative states [122,123,124,125], among others. Although these works share many common aspects, different methods were used, such as methods depending on a time-series model, methods not using any a priori model, and methods estimating effective connectivity [126]. Recent techniques and analytical methods have made it possible to study connectivity through the correlation level between the functional time series of regions during rest [126,127].

The investigation of the DMN continues to be an expanding field, and overcoming methodological barriers is essential to advancing the understanding of its function and clinical implications.

## 5. Default Mode Network Maturation

Default Mode Network maturation during childhood is influenced by various factors, including environmental adversities such as child abuse, neglect, abandonment, and exposure to stressful environments. The DMN plays a critical role in brain development, impacting cognitive and emotional processes throughout childhood and adolescence [25]. As the brain matures, the DMN facilitates the integration of sensory and cognitive information, contributing to the consolidation of autobiographical memory, self-reflection, and emotion processing [128]. This neural network is intrinsically linked to identity and self-awareness, enabling introspection and projection of thoughts about the past, present, and future [11,110].

Research consistently demonstrates that child abuse and neglect lead to significant alterations in DMN maturation. Children exposed to these adversities often exhibit abnormal patterns of functional connectivity within the DMN, including hypo- or hyperconnectivity between its constituent regions [129,130]. Persistent stress during early development can lead to enduring alterations in brain architecture and function, affecting the organization of the DMN and contributing to mental health challenges in adulthood.

Longitudinal research indicates that adults who have experienced childhood abuse are more likely to exhibit abnormalities in DMN connectivity. These alterations are associated with symptoms of psychiatric disorders, including depression and post-traumatic stress disorder [131]. Additionally, adverse childhood experiences can influence the expression of genes related to neuronal development and plasticity, further impacting DMN maturation [129]. Furthermore, abandonment and exposure to stressful environments have also been linked to changes in DMN maturation. Children raised in foster care or dysfunctional family contexts often lack emotional and social stability, potentially compromising proper DMN development [132]. Functional magnetic resonance imaging studies reveal that adults exposed to childhood stress exhibit altered DMN connectivity patterns, reflecting difficulties in emotional regulation and cognitive self-regulation [133].

Poverty and socioeconomic stress increase childhood exposure to adversity, impacting Default Mode Network connectivity [134]. Low-income children face neglect, food insecurity, and chronic stress, leading to altered connectivity between the medial prefrontal cortex and posterior cingulate gyrus, which affects self-regulation and memory [135,136]. Limited healthcare and psychological support further hinder cognitive and emotional development [137].

Cultural and environmental factors also shape DMN maturation. Societies tolerating harsh discipline or domestic violence expose children to early stress, impairing emotional regulation [129,132,138]. Weak social support and inadequate child protection policies exacerbate these effects [133].

Family structure plays a key role. Single-parent households and caregivers with psychiatric disorders increase the risk of DMN alterations [139]. Children from unstable homes show hyperconnectivity between the posterior cingulate gyrus and the precuneus, possibly as a stress adaptation [140].

Quality education and enriched environments offer protective effects. Early education programs enhance DMN connectivity, particularly in areas linked to executive function and emotional processing [141], whereas under-resourced schools heighten vulnerability to stress-related DMN dysfunction [142].

Chronic stress during childhood can lead to structural brain changes, including volume reductions in areas associated with the DMN, such as the medial prefrontal cortex and posterior cingulate cortex [143]. These structural changes can endure into adulthood, impacting mental health by contributing to mental health challenges such as anxiety, depression, interpersonal challenges, and overall psychological well-being [74,144]. Nevertheless, timely interventions and preventive strategies hold the potential to alter these dysfunctional neural connectivity patterns, fostering robust recovery and positive adaptation across the lifespan. Therapeutic approaches specifically targeting the DMN, including cognitive-behavioral therapy (CBT), exposure therapy, and occupational therapy, may play a pivotal role in restoring neural function and enhancing clinical outcomes for individuals who have experienced childhood adversity [11,145].

In contrast, enriched environments may have protective effects on DMN maturation in children exposed to adversity. Animal studies have shown that exposure to enriched environments, characterized by cognitive, social, and physical stimulation, can promote healthy brain development and modulate the functional connectivity of the DMN [146]. Early interventions based on enriched environments, such as high-quality early childhood education programs and psychosocial interventions, have been associated with improvements in executive function and emotional regulation, as well as positive changes in DMN connectivity in children exposed to adversity [147].

## 6. Default Mode Network and Social Relations

The process of understanding the world is dynamic and constantly evolving. It involves integrating information received with our existing knowledge across various time scales [148]. As social beings, humans spend significant time evaluating their own positions and social connections, as well as those of others. This mental activity encompasses self-reflection, consideration of others, and the exchange of ideas throughout our lives [149].

Dunbar and colleagues proposed the ‘social brain hypothesis’, which posits that the large brains observed in primates are adaptations to the computational demands of complex social systems within the primate order [150]. Consequently, our thoughts and actions in any given moment are influenced by three key factors: the current sensory input from the world; our recent memories that contextualize the present; and our long-term memories, beliefs, and emotions that shape our processing of incoming information [148].

Over the past two decades, extensive research has focused on the human social brain across various domains. These domains include understanding others, self-awareness, self-regulation, and the processes that occur at the interface between individuals and their social environment [151]. Social cognition, which specifically refers to the ability to comprehend other people’s emotional, mental, psychological, and behavioral states, has been a central topic of investigation [151]. Recent studies have highlighted the strong involvement of regions within the DMN in tasks related to understanding and interacting with others. These tasks include perceiving and interpreting emotional states, demonstrating empathy, and inferring beliefs and intentions [149,152].

Moreover, beyond the DMN regions, other brain areas also play crucial roles in social behaviors. These behaviors often require cognitive processes such as gathering, retrieving, and processing information about the lives, relationships, and mental states of others [59]. The study of human social relationships is a multidisciplinary field that has garnered increasing interest in cognitive neuroscience. Within this context, the DMN serves as an essential component for understanding social interactions. These brain regions display intrinsic activity either during an individual’s rest or when they are engaged in tasks unrelated to external stimuli [25,59,94].

DMN comprises several brain regions, including the posterior cingulate cortex, medial prefrontal cortex, precuneus, inferior parietal cortex, and medial temporal cortex [153].

Functional neuroimaging studies reveal that DMN activity correlates with the capacity to infer mental states and engage in empathetic processes [59,149]. Additionally, alterations in DMN connectivity have been observed in individuals with autism spectrum disorders (ASD), who often experience difficulties in understanding social interactions [154]. These findings emphasize the critical involvement of the DMN in the pathophysiology of ASD and offer valuable insights into the neural mechanisms that underlie social deficits in these disorders [154].

## 7. DMN Influencing Factors

The age factor significantly influences DMN. Research indicates that DMN undergoes both structural and functional changes across the lifespan, exhibiting distinct connectivity and activity patterns in children, adults, and the elderly [155]. These changes are likely associated with brain development and life experiences.

Apart from age, genetic factors also play a crucial role in modulating the DMN. Neuroimaging studies reveal that genetic variations impact the DMN’s structure and function, affecting its connectivity and resting-state activity [139]. These findings suggest that individual genetic factors contribute to variations in DMN organization.

Environmental factors also shape DMN activity. Stress, education, lifestyle, and exposure to environmental stimuli can all influence DMN function [139]. For instance, chronic stress alters DMN connectivity, impacting the brain’s ability to engage in complex cognitive and emotional processes. Furthermore, individual characteristics—such as personality traits and mental health history—also influence DMN activity [156]. For example, individuals with extroverted traits may exhibit distinct DMN connectivity patterns compared to those with introverted traits [156].

### 7.1. Age

The activity and connectivity of DMN have been extensively investigated in relation to age, revealing complex patterns of change throughout life. Longitudinal and cross-sectional studies have shown that DMN undergoes significant structural and functional changes during the course of aging. Early in life, the DMN appears to be in the process of development, with distinct patterns of connectivity and activity in children and adolescents compared to young adults [155]. These differences may reflect the maturation of the brain and the development of complex cognitive skills throughout childhood and adolescence.

As we move into adulthood and middle age, relative stability is observed in the connectivity patterns of the DMN, with some subtle modifications associated with normal aging. However, studies have also reported a reduction in functional connectivity within the DMN in older adults, along with greater individual variability in this population [54,157]. These changes may be related to neurodegenerative processes associated with aging, such as brain atrophy and volume loss in key regions of the DMN.

In addition, evidence suggests that age can modulate the response of the DMN to different cognitive and emotional tasks. For example, studies have shown that older adults tend to exhibit greater activation of the DMN during autobiographical memory tasks compared to young adults [156]. These findings suggest that DMN may play a compensatory role in preserving cognitive function in the face of aging and brain degeneration.

DMN is susceptible to modifications throughout the life course, with structural and functional changes associated with development, aging, and neurodegenerative processes. Understanding how these changes affect brain function is fundamental to elucidating the mechanisms underlying healthy aging and cognitive decline and can provide important insights for the development of therapeutic interventions in age-related disorders [158,159].

### 7.2. Emotional State

DMN activity and connectivity have been associated with a variety of emotional states, including fear, sadness, anger, happiness, and empathy. During fearful situations, the DMN may be involved in the cognitive evaluation and emotional reappraisal of aversive stimuli, influencing the activity of limbic regions associated with the generation and expression of fear [94]. This regulation of adaptive emotional responses may have significant implications for the understanding and treatment of fear-related disorders, such as anxiety disorder and post-traumatic stress disorder [160,161].

Functional neuroimaging studies indicate that the DMN maintains significant neural connections with the limbic system, including the amygdala and anterior cingulate cortex, regions crucial for emotional processing, including fear [83]. The interaction between DMN and the limbic system during fearful situations can influence the cognitive evaluation and emotional reappraisal of aversive stimuli [94]. For example, the DMN can modulate the activity of the amygdala, a central structure in the generation and expression of fear, affecting the individual’s interpretation and response to the threatening stimulus [162]. In addition, dysfunctions in the DMN have been associated with fear-related disorders such as anxiety disorder and post-traumatic stress disorder [162,163].

Similarly, sadness is associated with specific patterns of DMN activity, with greater activation of DMN regions during states of sadness and rumination [164]. This hyperactivity of the DMN during states of sadness may contribute to the perseverance of negative thoughts and unpleasant emotions, characteristic of depressive disorders [25,165].

The interaction between the DMN and other brain regions, such as the anterior cingulate cortex and the amygdala, can modulate adaptive emotional responses and influence the expression of sadness. Dysfunctions in the DMN have been associated with sadness-related disorders such as depression and mood disorder [15,166]. Therapeutic approaches that aim to modulate the activity of the DMN, such as CBT and exposure therapy, may represent promising strategies for the treatment of these emotional disorders related to sadness [162,163].

On the other hand, anger is associated with a decrease in functional connectivity within the DMN and an increase in the activity of brain regions involved in emotional regulation, such as the dorsolateral prefrontal cortex [167]. These changes in DMN activity may reflect a reorientation of neural resources to process and regulate intense emotional states, such as anger. Dysfunctions in DMN have been associated with anger-related disorders such as impulse control disorders and intermittent explosive disorder [162,163].

In addition, the DMN also plays a role in regulating emotional states such as happiness. A group-independent component analysis (ICA) revealed that intensified functional connectivity of the DMN was correlated with lower levels of contentment. Compared to happier individuals, those who expressed greater discontent showed greater functional connectivity in the central regions of the DMN, such as the MPFC, PCC, and IPL [168]. In addition, the intensity of functional connectivity in these regions was positively related to scores on two subscales of rumination: incubation and reflection [169,170,171]. The increased functional connectivity within the DMN among unhappy individuals may indicate a tendency toward excessively negative self-reflection. These current results reinforce the idea that unhappiness is linked to the hyperconnectivity of DMN areas in a non-clinical sample and that the intensity of this hyperconnectivity between the MPFC, PCC, and IPL is correlated with the rumination trait [172,173]. Unhappy individuals may devote more time to rumination about negative feelings, thoughts, and emotions [21,54,174,175,176,177,178]. The DMN activates when people are engaged in processes of unrestricted self-reflection and deactivates during goal-oriented activities.

The interaction between DMN and other brain networks, such as the attention network and the Salience Network, can modulate a person’s ability to experience and sustain positive emotional states, such as happiness [93,179]. Dysfunctions in DMN have been associated with mood disorders, such as depression, which are often related to deficits in the ability to experience happiness and pleasure [85,165].

Relations between the Default Mode Network and empathy are a field of study in social neuroscience. The DMN, an intrinsic neural network of the brain, plays a central role in introspection, self-awareness, and theory of mind, as well as in cognitive abilities that are essential for empathy [3]. Empathy, an essential component of human social cognition, refers to the ability to understand and share the emotional states of other individuals [180,181]. This ability plays a crucial role in social interaction, allowing people to effectively recognize and interpret the emotional state of their peers [182]. Functional neuroimaging studies suggest that the DMN is involved in processing social information and in the ability to understand and share other people’s mental states, which are fundamental aspects of empathy [59,152]. During social interactions and tasks that require empathy, the DMN shows greater activity, indicating its role in forming mental representations of other people’s emotional states [183,184].

A number of studies have shown DMN hyperactivity at rest in individuals with anxiety disorders, such as Generalized Anxiety Disorder (GAD), panic disorder, and post-traumatic stress disorder (PTSD) [94,185,186]. For example, using fMRI, Rauch et al. symptoms [14] observed that patients with GAD show greater DMN activity during cognitive tasks, suggesting difficulty in deactivating this network during distracting activities [187]. This hyperactivity can contribute to excessive attention to worrying thoughts and difficulty in directing attention to external stimuli, perpetuating anxious symptoms [14].

Studies have explored the mechanisms by which the DMN can influence anxiety. Emotional regulation is a key function of the DMN, and its dysregulation can amplify negative emotional responses common in anxiety disorders [14]. For example, Etkin and Schatzberg [188] suggested that DMN facilitates rumination about past and future worries associated with anxious symptoms [188]. Abnormal patterns of connectivity within the DMN are also correlated with rumination and excessive worry [3].

Understanding this relationship is crucial to developing effective therapeutic approaches. Interventions designed to modulate DMN activity, such as mindfulness meditation and cognitive-behavioral therapy, show promise in the treatment of anxiety disorders, aiming to regulate rumination and excessive worry [14]. It is worth noting that other dysregulations in DMN are associated with conditions such as autism, schizophrenia, and depression [189].

Structural neuroimaging studies reveal alterations in the brain morphology of individuals with anxiety disorders, including reductions in the volume of areas associated with DMN, such as the medial prefrontal cortex and posterior cingulate [144]. These structural changes can hinder emotional processing and cognitive regulation.

Acute anxiety can temporarily dysregulate the DMN, resulting in increased activity in regions associated with self-criticism and excessive worry [188], interfering with the alternation between states of rest and activity. Chronic exposure to anxiety can lead to changes in the functional connectivity of the DMN over time, predisposing individuals to psychiatric disorders [131]. Chronic anxiety contributes to long-lasting changes in the structure and function of the DMN, increasing the risk of mental health problems throughout life. These findings highlight the importance of understanding the interactions between MDD and anxiety in order to improve the therapeutic approach to these disorders [190].

The interaction between DMN and other brain networks, such as the attention mode network and the executive control network, can modulate a person’s ability to put themselves in another person’s shoes and respond in an empathetic way [182]. Dysfunctions in the DMN have been associated with empathic difficulties and deficits in the ability to understand and respond to other people’s emotional states, common features in autism spectrum disorders and other empathy-related disorders [152,191].

### 7.3. Cognitive States

DMN plays an important role in mental foresight, i.e., the ability to anticipate and plan for the future. Studies have shown that the DMN is active during the simulation of future events and in the elaboration of hypothetical scenarios, suggesting that this network is involved in the construction of expectations and decision-making [94]. Social processing is also closely linked to DMN activity. Research has shown that the DMN is involved in attributing mental states to others, such as beliefs, intentions, and emotions, and in navigating complex social interactions [59]. This ability to understand and anticipate the thoughts and feelings of others is fundamental to social functioning and empathy.

Furthermore, the DMN plays a role in regulating the attention and focus of the mind. Studies have shown that DMN activity is inversely related to the activity of neural networks involved in task-oriented attention, suggesting that the DMN may be involved in deactivating external cognitive processes during periods of introspection and reflection [55]. DMN plays a crucial role in regulating a variety of cognitive states, including introspection, autobiographical memory, mental prospection, social processing, and attention regulation [71,94]. Understanding how the DMN is involved in the dynamics of these cognitive states can provide important insights into normal brain functioning and the development of therapeutic interventions in neurological and psychiatric disorders [69,192,193].

### 7.4. Sensory Experiences

Although initially associated mainly with internal cognitive processes, recent studies have highlighted its contribution to sensory perception and the modulation of states of consciousness. During sensory experiences such as vision, hearing, touch, and smell, the DMN displays a complex dynamic of activity that reflects its interaction with other neural networks involved in sensory processing. Research using functional neuroimaging techniques, such as fMRI, has revealed that the DMN is active during periods of rest or in states of reduced attention but can also be recruited during the conscious perception of sensory stimuli [194].

The integration of sensory information by the DMN is crucial for the formation of conscious perceptions and the attribution of meaning to environmental stimuli. For example, during the visualization of a complex scene, the DMN may be involved in the retrieval of memories associated with the objects present in the scene, the elaboration of internal narratives about the situation, and the emotional evaluation of visual stimuli [71].

In addition, the DMN plays a role in regulating attention and sensory processing during states of rest and reduced attention. Although it is often considered an “inactive” network during sensory perception, the DMN can play a role in deactivating external cognitive processes and facilitating attention to internal stimuli [55,194].

In summary, the DMN plays a multifaceted role in the integration and processing of sensory experiences. Its dynamic activity during periods of rest and sensory perception suggests that this network plays a central role in the construction of conscious experience and the regulation of states of consciousness.

### 7.5. Social Experiences

DMN is highly sensitive to social experiences. Recent studies have shown that social interactions can have a significant impact on the activity and connectivity of the DMN, influencing its function and plasticity over time [193]. Positive social experiences, such as affectionate interactions, social support, and meaningful emotional connections, tend to modulate DMN activity in a positive way [192,193]. During these interactions, the DMN can show an increase in functional connectivity between its constituent regions, reflecting the intensification of self-reflection and empathy [59]. For example, in situations of emotional sharing, such as an intimate conversation with a close friend, the DMN can activate the process of one’s own and the other’s emotions, promoting greater mutual understanding and a sense of emotional connection [149,152].

Instead, negative social experiences, such as interpersonal conflicts, social isolation, and rejection, can lead to changes in DMN activity. During situations of social stress, the DMN may show a decrease in functional connectivity and greater activation of regions associated with emotional regulation, reflecting an adaptive response to the perceived social threat [59]. In addition, experiences of social exclusion or ostracism can lead to a dysregulation of the DMN, impairing the capacity for self-reflection and empathy and increasing the risk of mental health problems such as depression and anxiety [195].

The plasticity of the DMN in response to social experiences suggests that this neural network is highly malleable and susceptible to environmental influences. Therefore, intervention strategies aimed at promoting positive social interactions and reducing social stress may have the potential to modulate DMN activity and improve emotional and mental well-being [59,196,197].

Thus, social experiences have a significant impact on the activity and connectivity of the DMN, influencing its function and plasticity over time. Understanding how social interactions shape DMN functioning can provide valuable insights into the mechanisms underlying social and emotional processing and inform intervention strategies to promote emotional and mental well-being [198].

### 7.6. States of Consciousness

During periods of wakefulness and rest, the DMN exhibits increased activity, suggesting its participation in introspective, self-reflective, and self-referential processes [71]. This increase in DMN activity during the resting state may be associated with the wandering mind, self-reflection, and the retrieval of personal memories [94]. During sleep, the DMN tends to show reduced activity, especially during REM sleep [199,200]. This reduction in DMN activity during sleep may be related to the temporary disconnection of neural networks associated with self-reflection and introspection, thus allowing memory consolidation and brain restoration processes to take place [201]. In addition, the DMN has also been implicated in altered states of consciousness, such as meditation and psychedelic experiences. During meditation, for example, a reduction in DMN activity is observed, suggesting a decrease in mental rumination and greater mindfulness of the present moment [122]. During psychedelic experiences, such as ingesting psychedelics, there is a temporary disorganization of the DMN, which may be associated with changes in the perception of the self and the dissolution of the ego [202].

### 7.7. Chronic Stress

Chronic stress in childhood can have a significant impact on the formation and maturation of myelin, the protective sheath that surrounds neuronal axons [203]. Myelin is essential for the efficient transmission of neural signals and for the structural integrity of neurons [204]. Alterations in the formation and maintenance of myelin can have important consequences for brain function in the short, medium, and long term [203,205]. Exposure to chronic stress during childhood can trigger neurobiological responses that directly affect myelin formation [138]. Cortisol, a hormone released in response to stress, can have adverse effects on the glial cells responsible for producing and maintaining myelin, such as oligodendrocytes [203]. Studies in animal models have shown that high levels of cortisol can inhibit the proliferation and differentiation of oligodendrocytes, resulting in a reduction in myelin production [138,203,204].

These changes in myelination can have an immediate impact on information processing and the transmission of neural signals [203]. Myelin acts as an electrical insulator around axons, allowing nerve impulses to be conducted quickly and efficiently [101]. When myelin is compromised, the transmission of neural signals can be impaired, leading to delays in communication between nerve cells and compromising the proper functioning of neural networks [205].

The dysfunction in myelination can contribute to cognitive, emotional, and behavioral difficulties in the medium term. Studies have associated alterations in myelin with impairments in memory, emotional processing, attention, and cognitive control [206]. In addition, reduced myelin integrity may increase vulnerability to the development of psychiatric disorders, such as depression and anxiety, in adolescence and adulthood [128].

The effects of chronic stress on myelination can manifest as lasting structural and functional changes in the brain in the long term [101]. Longitudinal studies have shown that individuals exposed to adversity in childhood show differences in the integrity of the cerebral white matter, reflecting impairment in myelination and neural connectivity [207]. These alterations can persist throughout life and be associated with a greater risk of mental and cognitive health problems in adulthood [207]. In summary, chronic stress in childhood can interfere with the formation and maturation of myelin, with adverse impacts on brain function in the short, medium, and long term. These alterations can contribute to a series of cognitive, emotional, and behavioral difficulties, increasing the risk of developing psychiatric disorders throughout life [204].

DMN modulation throughout life, from infancy to adulthood, is influenced by a variety of factors, including childhood adversity and chronic stress [208,209]. These modulations can have significant consequences for brain and mental health, increasing the risk of neurological problems in adulthood [210]. Understanding these modulations is crucial for developing early intervention strategies and effective treatments for a variety of neurological conditions.

In adulthood, DMN modulations continue to play an important role in brain and mental health. Longitudinal studies have shown that adults who have been victims of childhood abuse are more likely to have abnormalities in DMN connectivity associated with symptoms of psychiatric disorders such as depression and post-traumatic stress disorder [131]. In addition, chronic stress throughout adulthood can lead to structural changes in the brain, including reductions in the volume of areas associated with DMN, increasing the risk of mental health problems such as anxiety and depression [144].

## 8. Medical and Psychological Diagnoses Associated with Default Mode Network

Abnormal modulations of DMN have also been associated with a variety of neurological problems. DMN changes have been observed in patients with autism spectrum disorder (ASD), with hyperconnectivity in some regions and hypocoherence in others [154]. In addition, the DMN also plays a role in Alzheimer’s disease (AD), with studies showing early dysfunction in this network in the early stages of the disease [7]. The most fundamental idea for using functional connectivity in pathologies is to use the intensity of functionally coupled correlations as markers of the integrity of the brain system [7]. The following topics are some examples of the use of brain functional network parameters in the study of some neurological diseases, aging, and surgical planning.

### 8.1. Alzheimer’s Disease

Alzheimer’s disease is a neurodegenerative condition characterized by progressive memory loss, cognitive impairment, and behavioral changes [211,212]. It affects millions of people worldwide and represents one of the main causes of dementia in the elderly population. One of the most active areas of study in Alzheimer’s research is the investigation of the underlying brain changes that occur during the course of the disease [111]. Functional neuroimaging studies have consistently shown changes in the activity and connectivity of the DMN in AD patients. For example, a recent review of functional magnetic resonance imaging studies highlighted that AD patients often exhibit a decrease in functional connectivity within the DMN and an increase in functional connectivity between the DMN and other brain networks, such as the executive control network [213].

These changes in DMN connectivity have been associated with specific clinical symptoms of AD, such as episodic memory deficits and changes in executive function. In addition, longitudinal studies have suggested that changes in DMN connectivity can occur in the early stages of the disease, even before the onset of clinical symptoms [214].

Recent research has also explored the role of specific characteristics of the DMN, such as its temporal variability and its organization into subnetworks, in AD. For example, a recent study using temporal variability analysis of the DMN found that AD patients show a reduction in the temporal variability of the DMN compared to healthy controls, and this reduction was associated with greater cognitive impairment [215].

Furthermore, the identification of DMN-specific imaging markers, such as measures of functional and structural connectivity, may provide additional insights into the progression of AD and aid in the development of biomarkers for early diagnosis and monitoring of the disease [216]. Reduced clustering and increased path length indicate small-world network degradation, although the results are not consensual [217]. Greicius showed that functional connectivity in the DMN in Alzheimer’s patients is disrupted when compared to normal controls [7].

### 8.2. Schizophrenia

Schizophrenia is a complex psychiatric condition characterized by a variety of symptoms, including hallucinations, delusions, disorganized speech, and cognitive deficits [218]. In recent years, there has been a growing interest in understanding how alterations in the Default Mode Network may be associated with the pathophysiology of schizophrenia [219].

Functional neuroimaging studies have provided significant insights into alterations in DMN in patients with schizophrenia. For example, a recent systematic review of fMRI studies highlighted hypoconnectivity within the DMN in patients with schizophrenia, especially in regions such as the medial prefrontal cortex and posterior cingulate cortex [220].

In addition, research has explored the relationship between abnormalities in the DMN and the specific symptoms of schizophrenia. For example, a high-resolution neuroimaging study identified a reduction in functional connectivity between the dorsolateral prefrontal cortex and the posterior cingulate cortex in patients with schizophrenia, correlating with symptoms of disorganization and cognition [221].

Other studies have investigated how abnormalities in DMN can affect other brain networks and contribute to the symptoms of schizophrenia. For example, recent research using network connectivity techniques has identified dysfunctions in the coupling between the DMN and the executive control network in patients with schizophrenia, suggesting a neurobiological basis for the executive cognition deficits observed in the disease [222].

Individuals with schizophrenia often show alterations in the connectivity of DMN, including hypoconnectivity between the medial prefrontal cortex and other regions of the DMN. These alterations can contribute to symptoms such as cognitive dysfunction and altered perception of reality [65,223]. These advances in neuroscientific research have the potential to inform the development of new therapeutic approaches for schizophrenia aimed at modulating the DMN and its interactions with other brain networks.

### 8.3. Anxiety Disorders

Anxiety is a common psychiatric disorder characterized by excessive worry, persistent fear, and physical symptoms such as palpitations and sweating [218]. Functional neuroimaging studies have consistently shown alterations in the activity and connectivity of the DMN in individuals with anxiety disorders [224]. For example, a recent systematic review of functional magnetic resonance imaging studies highlighted the hyperconnectivity of the DMN in patients with GAD during rest and emotional processing tasks [225]. Changes in DMN activity have been observed in patients with anxiety disorders, such as GAD, panic disorder, and specific phobias. These changes usually involve hyperactivity of the DMN, especially during states of rumination or excessive worry [84].

Several researchers have investigated how changes in the DMN can contribute to anxiety symptoms. For example, a longitudinal study showed that DMN hyperconnectivity in adolescents was associated with an increased risk of developing anxiety over time [84]. Other studies have explored the effects of therapeutic interventions on DMN connectivity in patients with anxiety. For example, cognitive-behavioral therapy has been shown to reduce DMN hyperconnectivity in patients with social anxiety disorder, correlating with improved anxiety symptoms [60]. These findings suggest that DMN plays an important role in the pathophysiology of anxiety and may represent a promising therapeutic target for the development of new treatment approaches.

### 8.4. Post-Traumatic Stress Disorder (PTSD)

The relationship between post-traumatic stress disorder (PTSD) and the DMN has been widely investigated due to its clinical relevance and potential to provide insights into the neurobiology of PTSD [226]. DMN, an intrinsically connected neural network that plays a central role in self-reflection, autobiographical memory, and emotional processing, has been the target of study in PTSD patients to understand the neurofunctional alterations associated with this disorder [226].

Individuals with a diagnosis of PTSD often have dysfunctions in the DMN, including hyperactivity in regions associated with rumination and the reactivation of traumatic memories [227]. These alterations can contribute to symptoms such as flashbacks and avoidance of trauma-related thoughts [227,228]. Functional neuroimaging studies, such as fMRI and electroencephalography (EEG), have consistently demonstrated dysfunction in the DMN in individuals with a diagnosis of PTSD [229]. For example, a recent functional connectivity analysis showed hypoconnectivity between the medial prefrontal cortex and the posterior cingulate cortex, key regions of the DMN, in individuals with a diagnosis of PTSD [230]. This hypoconnectivity may be associated with difficulties in emotional regulation and the integration of trauma-related information [227,228].

Studies have explored how alterations in the DMN are related to specific PTSD symptoms [227,231]. For example, it has been observed that hypoconnectivity between the medial prefrontal cortex and the posterior cingulate cortex correlates with symptoms of avoidance and re-experiencing trauma [232]. These findings suggest that specific dysfunctions in the DMN may be linked to different aspects of PTSD and may contribute to the heterogeneity of symptoms observed in this disorder.

Therapeutic interventions, such as eye movement desensitization and reprocessing (EMDR) therapy, have been shown to modulate DMN activity in individuals with a diagnosis of PTSD [227]. Studies have shown that EMDR can reduce DMN hyperactivity, especially in the medial prefrontal cortex, correlating with an improvement in PTSD symptoms. This suggests that DMN regulation may be an important mechanism of action for effective therapeutic interventions in PTSD. Investigating the relationship between DMN and PTSD provides valuable insights into the neurobiological mechanisms underlying this complex disorder. Understanding alterations in the DMN may pave the way for the development of new therapeutic approaches for PTSD aimed at modulating DMN activity and restoring brain function in patients affected by this debilitating disorder.

### 8.5. Depression

The relationship between depression and DMN has been widely investigated due to its importance in understanding the neurobiology of depression and in developing new therapeutic approaches [165]. Depression is associated with a variety of alterations in DMN activity, including hyperconnectivity between DMN and limbic regions such as the posterior cingulate cortex and hippocampus. These changes can contribute to symptoms such as rumination, self-criticism, and recurrent negative thoughts [165].

Functional neuroimaging studies, such as fMRI and electroencephalography (EEG), have consistently shown alterations in DMN activity and connectivity in patients with depression [233]. For example, a recent meta-analysis demonstrated hyperconnectivity between regions of DMN, such as the medial prefrontal cortex and the posterior cingulate cortex, in patients with depression [234]. This hyperconnectivity may be associated with rumination, and negative self-reference thought patterns observed in depression.

Studies have explored how alterations in DMN are related to specific symptoms of depression [15,123]. For example, hyperconnectivity between regions of DMN has been found to correlate with the severity of depressive symptoms, including anhedonia and feelings of hopelessness [235]. These findings suggest that specific dysfunctions in the DMN may be associated with different aspects of depression and may contribute to the heterogeneity of symptoms observed in this disorder.

Therapeutic interventions, such as CBT and transcranial magnetic stimulation (TMS), have been investigated for their effects on DMN activity in patients with depression [123]. Studies have shown that CBT can modulate DMN activity, reducing hyperconnectivity between its regions and correlating with an improvement in depressive symptoms [1]. In addition, TMS has been associated with changes in DMN connectivity, especially in the medial prefrontal cortex, suggesting that DMN regulation may be an important mechanism of action for therapeutic interventions in depression [212].

### 8.6. Attention Deficit Hyperactivity Disorder (ADHD)

Attention deficit hyperactivity disorder (ADHD) is characterized by persistent attention difficulties, hyperactivity, and impulsivity, affecting the cognitive and behavioral functioning of those affected [236]. The relationship between ADHD and the DMN has been the subject of study in order to understand the neurobiological basis of this disorder and identify possible biomarkers for diagnosis and intervention [173]. Patients with ADHD often have dysfunctions in DMN activity, including hypoactivity in regions associated with attention control and behavior regulation. These alterations can contribute to symptoms such as difficulty concentrating and impulsivity [148,236].

Functional neuroimaging studies have shown differences in the activity and connectivity of the DMN in individuals with ADHD compared to healthy controls. For example, a recent meta-analysis showed that children with ADHD have hypoconnectivity in regions of the DMN, such as the medial prefrontal cortex and posterior cingulate cortex, during rest [14]. This hypoconnectivity may be related to difficulties in self-reflection and in processing autobiographical memories, which are functions associated with the DMN. In addition, studies have explored how alterations in the DMN are related to specific ADHD symptoms. For example, it has been observed that hypoconnectivity in regions of DMN correlates with the severity of symptoms of inattention and impulsivity in children with ADHD [237]. These findings suggest that specific dysfunctions in the DMN may contribute to the different clinical aspects observed in this disorder.

Therapeutic interventions, such as behavioral and pharmacological therapy, have been investigated for their effects on DMN activity in patients with ADHD [236,238]. Studies have shown that behavioral therapy can modulate the connectivity of DMN, restoring normal patterns of connectivity between its regions and correlating with an improvement in symptoms of inattention and hyperactivity [227,239]. In addition, pharmacological therapy, especially with stimulants such as methylphenidate, has been associated with modifications in DMN activity, suggesting that DMN regulation may be an important therapeutic target for the treatment of ADHD [240].

These are just some of the psychiatric diagnoses associated with DMN, and research continues to explore the complex interactions between DMN activity and the pathophysiology of these conditions [173,237]. Understanding these associations can help in the development of new therapeutic approaches and the refinement of diagnostic strategies for these diseases. Another class of applications in surgical planning [238]. Treatments for brain tumors can involve the removal of brain tissue, and mapping the locations of brain systems is critical information that allows the surgeon to maximize the size of the resection while minimizing damage to the eloquent cortex [241].

## 9. Neuroplasticity and DMN

The brain’s ability to reorganize itself structurally and functionally in response to experiences, cognitive training, and therapeutic interventions has been associated with significant changes in DMN, as evidenced by various studies in the field [192,242].

Research using functional neuroimaging techniques, such as fMRI, has revealed changes in the activity and connectivity of the DMN in response to interventions that promote neuroplasticity. For example, a study by Brewer et al. (2011) investigated the effects of practicing mindfulness meditation on DMN in human participants [122]. The results showed that regular meditation practice was associated with a reduction in resting DMN activity, indicating a greater ability to switch off the mind and focus on the present moment.

Other studies have reported changes in functional connectivity within the DMN in response to mindfulness meditation, suggesting a reorganization of the underlying neural network [122,243]. These findings highlight the ability of meditation practice to promote neuroplasticity in the DMN, resulting in functional changes that may be associated with improvements in emotional regulation, attention, and mental well-being.

In addition to mindfulness meditation, therapeutic interventions for psychiatric diagnoses have also been associated with changes in the DMN in response to neuroplasticity. For example, a study by Goldin and Gross (2010) investigated the effects of CBT on DMN in patients with social anxiety disorder [244]. The results showed that CBT was related to a normalization of DMN activity and a reduction in rumination, a common symptom of this disorder.

These studies highlight the ability of the DMN to reorganize itself in response to interventions that promote neuroplasticity, providing important insights into the neural mechanisms underlying these changes [245,246]. A better understanding of how neuroplasticity affects DMN could lead to the development of more effective therapeutic interventions for a variety of neuropsychiatric conditions, thus improving the quality of life of affected individuals [245,246].

## 10. Final Considerations

The literature review on the DMN has offered a thorough understanding of this neural network and its involvement in various cognitive, emotional, and behavioral processes. The DMN is fundamentally linked to introspection, self-reflection, and the creation of mental narratives about oneself and the surrounding world, playing a vital role in the consolidation of episodic memory, emotional processing, and cognitive regulation. Research on DMN has extended beyond humans to include non-human animals, providing valuable insights into the evolution and adaptive functions of this neural network. Additionally, studies on DMN’s maturation during childhood have underscored the significant impact of environmental factors, such as child abuse, neglect, and chronic stress, on the organization and function of this network. These findings have important implications for healthy brain development and the risk of mental health issues throughout life. The connection between DMN and medical and psychological diagnoses like anxiety has been investigated, revealing abnormal activity and connectivity patterns associated with these conditions. Gaining a deeper understanding of these interactions could pave the way for more effective therapeutic approaches.

Studying changes in DMN activity in pathological conditions may provide new therapeutic insights. For example, investigations into how DMN is modulated during neurofeedback interventions or cognitive therapies could open new avenues for personalized treatments of emotional and cognitive disorders. Another promising area of research involves the relationship between DMN and brain plasticity, particularly in the context of neuroplasticity induced by learning or experience. Understanding how the DMN adapts over time in response to environmental or therapeutic changes could reveal new aspects of brain resilience and recovery. Moreover, the DMN’s capacity to reorganize itself through neuroplasticity has been emphasized, presenting promising opportunities for developing interventions aimed at modulating this neural network to enhance brain and mental health.

In summary, studying the DMN is crucial for understanding the neural mechanisms behind cognition, emotion, and behavior, as well as for creating intervention and treatment strategies for various neuropsychiatric conditions. This neural network continues to be a captivating and significant research focus, with the potential to revolutionize our understanding of the human mind and promote mental well-being. Future research should focus on more dynamic and interactive approaches, considering brain plasticity and the interaction between neural networks in normal and pathological states.

## 11. Conclusions

The literature on the Default Mode Network has provided extensive insights into its critical role in shaping cognitive, emotional, and behavioral processes. The DMN’s association with introspection, self-reflection, and mental narrative construction underscores its importance in memory consolidation, emotional regulation, and cognitive control. Research extending beyond humans to non-human animals has enriched our understanding of its evolutionary and adaptive significance, while studies on childhood maturation have emphasized the profound influence of environmental factors like stress and neglect on DMN functionality.

The exploration of DMN activity in pathological conditions, including anxiety and mental health disorders, highlights its potential as a target for therapeutic advancements. Furthermore, its modulation through neurofeedback, cognitive therapies, and experiences of neuroplasticity reveals the network’s dynamic adaptability. These findings not only deepen our understanding of brain resilience but also present promising opportunities for the development of innovative, personalized interventions. As a cornerstone in the study of neural mechanisms, the DMN offers transformative possibilities for neuropsychiatric treatment and mental well-being. Future research should prioritize its interplay with other neural networks and its capacity for adaptation under changing environmental and therapeutic contexts.

## Figures and Tables

**Figure 1 biology-14-00395-f001:**
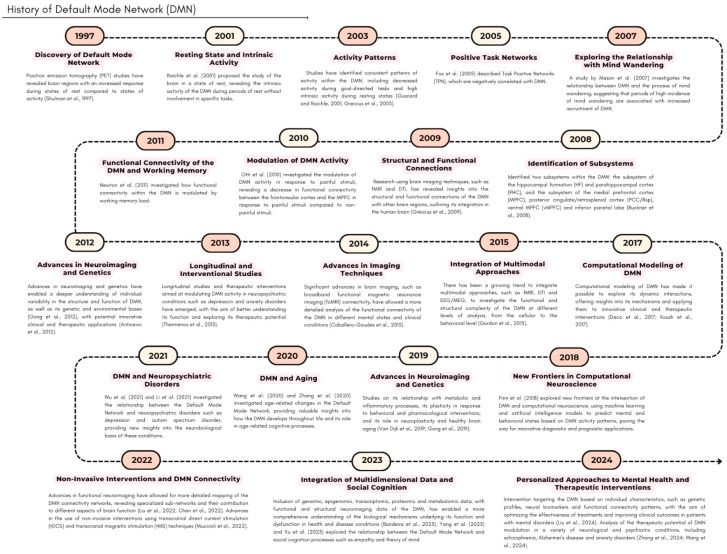
Timeline of the Default Mode Network’s history [16,17,18,19,20,21,22,23,24,25,26,27,28,29,30,31,32,33,34,35,36,37,38,39,40,41,42,43,44,45,46,47,48].

**Figure 2 biology-14-00395-f002:**
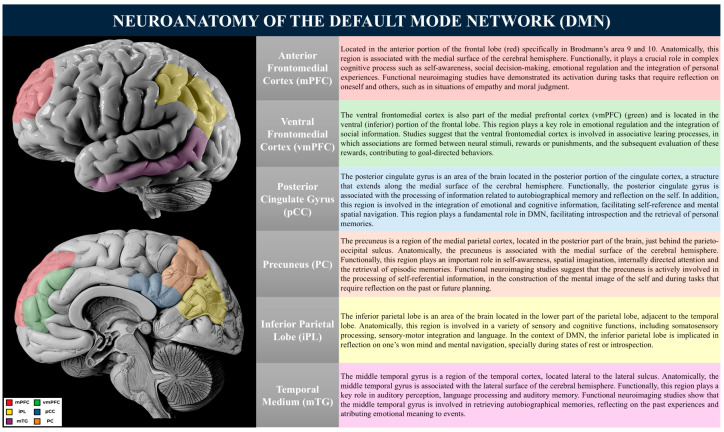
Default Mode Network neuroanatomy.

**Table 1 biology-14-00395-t001:** The main areas of investigation about the Default Mode Network.

Lines of Study and Research on DMN	
Basal Activation for Healthy Function	Findings have led to the hypothesis that DMN activity and connectivity play a crucial role in “offline” intrinsic activity, which is necessary to maintain balanced and stable internal states during periods of rest [4,25].
Reveries	It has been observed that DMN activity increases during periods of “daydreaming” and that an individual’s propensity to mental wandering is correlated with the DMN response [21]. In addition, the content of daydreams can modulate the activity and connectivity of the DMN, varying according to the different components of daydreams [54].
Autobiographical Memory	Several neuroimaging studies have shown that DMN activity and connectivity are modulated by tasks that require the retrieval of past events and by spontaneous autobiographical memory [55]. It has been suggested that autobiographical memory is one of the prominent processes showing the involvement of the DMN in mental representations generated using intrinsic information, independently of extrinsic information [56,57].
Prospective Memory	Memory “for the future”, like memory for the past, activates the DMN and shapes its connectivity [56,57,58].
Social Cognition	Many studies have found extensive overlap between the DMN and regions involved in social cognition, collectively known as the “social brain” [59]. In addition to being involved in self-referential processes, such as thinking about one’s own mental states, the DMN is involved in thinking about other people’s beliefs, intentions, and motivations and in preparing the intentional stance [60]. Importantly, thinking about other people is not confined to moments of disengagement but is also, and perhaps primarily, evident when people are involved with other social agents in the real world [21].
Meeting of the Intrinsic and Extrinsic	Taken together, the studies described here suggest that, on the one hand, the DMN is active during internally related thoughts that are generated by the individual themselves in the absence of external stimuli, as in the case of daydreaming, and can therefore be considered intrinsic, and, on the other hand, the DMN response is linked to external stimuli, especially during social interactions, and can therefore be considered extrinsic [61]. Thus, in this perspective, we combine many of these findings to suggest that the DMN is at the center of the interaction between the external and internal worlds [62].

**Table 2 biology-14-00395-t002:** The main tools for studying Default Mode Network.

Technique	Advantages	Limitations
Functional Magnetic Resonance Imaging (fMRI)	- High spatial resolution (millimeters).	- Limited temporal resolution (seconds).
	- Real-time brain activity mapping.	- Sensitive to artifacts (movement).
	- Non-invasive technique.	- Hemodynamic response may not directly reflect neuronal activity.
Diffusion Tensor Imaging (DTI)	- Visualizes structural brain connections (fiber tracts).	- Does not measure functional activity directly.
	- Studies white matter integrity.	- Sensitive to artifacts and noise.
		- Limited spatial resolution for fine tracts.
Structural Magnetic Resonance Imaging (sMRI)	- High spatial resolution and anatomical detail.	- Does not provide functional activity or connectivity data.
	- Non-invasive and widely used technique.	- Can be affected by patient movement during acquisition.
Magnetoencephalography (MEG)	- High temporal resolution (milliseconds).	- Low spatial resolution.
	- Sensitive to superficial cortical activity.	- Sensitive to noise and physiological artifacts.
Electroencephalography (EEG)	- Excellent temporal resolution (milliseconds).	- Limited spatial resolution.
	- Low cost and widely accessible.	- Sensitive primarily to superficial cortical activity.
Positron Emission Tomography (PET)	- Measures brain activity through radioactive substances.	- Low temporal resolution (seconds).
	- Observes metabolic processes and blood flow.	- Requires radioactive substances (invasive).
		- High cost and specialized equipment.

## Data Availability

No new data were created or analyzed in this study. Data sharing is not applicable to this article.

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
