# Peer review of "The Journey of the Default Mode Network: Development, Function, and Impact on Mental Health"

_biology, 2025, doi:10.3390/biology14040395_

Round 1

Reviewer 1 Report

Comments and Suggestions for Authors

The paper provides an extensive overview of the Default Mode Network (DMN), discussing its structural and functional properties, its development across the lifespan, and its implications for mental health. The authors effectively synthesize existing literature on the topic and highlight several key theories explaining DMN function. However, some areas of the manuscript could benefit from further elaboration, particularly in relation to the dynamic interplay between the DMN and other large-scale networks in tracking affective states across different timescales. Below are my comments:

  1. While the manuscript discusses numerous findings related to DMN function, it does not sufficiently critique the methodological constraints of fMRI studies, such as low temporal resolution, susceptibility artifacts, and inter-individual variability. Addressing these issues would enhance the discussion of current research limitations and future directions.
  2. The paper could be improved by incorporating recent research on how DMN connectivity is not static but dynamically shifts over time in response to changing emotional and cognitive demands. In this context, a recent study demonstrates how large-scale brain networks, including the DMN, exhibit time-varying connectivity patterns that align with ongoing affective states (Lettieri, G., Handjaras, G., Setti, F., Cappello, E. M., Bruno, V., Diano, M., ... & Cecchetti, L. (2022). Default and control network connectivity dynamics track the stream of affect at multiple timescales. Social cognitive and affective neuroscience, 17(5), 461-469.). Integrating this perspective would provide a more nuanced discussion of how DMN interactions evolve in real-time rather than being constrained to static connectivity descriptions.
  3. The manuscript could be strengthened by incorporating more discussion on the presence of DMN-like activity in non-human species. Comparative neuroscience studies have observed resting-state networks in animals, suggesting evolutionary continuity. A discussion of such studies would provide valuable context for the claims made about DMN function and development.

Overall, this manuscript is a valuable contribution to the literature on DMN function and its relevance to mental health. However, integrating critiques of methodological limitations, cross-species comparisons, and multimodal evidence would strengthen the discussion. Addressing these areas would provide a more balanced and comprehensive evaluation of the DMN’s role in cognition and psychopathology.

Author Response

March 3rd, 2025.

Editors-in-Chief

Biology

Dear Editor,

Please find enclosed the revised version of our manuscript entitled “The Journey of the Default Mode Network: Development, Function and Impact on Mental Health” submitted to Biology.

We addressed all of the reviewer’s comments, which were extremely relevant for improving our manuscript. A point-by-point answer is below addressed for each reviewer in separated letters. The changes asked by Reviewer 1 are highlighted in YELLOW in the manuscript.

Reviewer #1: “However, some areas of the manuscript could benefit from further elaboration, particularly in relation to the dynamic interplay between the DMN and other large-scale networks in tracking affective states across different timescales.”

Answer: Thank you for the suggestions. We added in the text more information about this topic.

Reviewer #1: “While the manuscript discusses numerous findings related to DMN function, it does not sufficiently critique the methodological constraints of fMRI studies, such as low temporal resolution, susceptibility artifacts, and inter-individual variability. Addressing these issues would enhance the discussion of current research limitations and future directions. “

Answer: Thank you for the suggestions. We added in the text more information about this topic.

Reviewer #1: “The paper could be improved by incorporating recent research on how DMN connectivity is not static but dynamically shifts over time in response to changing emotional and cognitive demands. In this context, a recent study demonstrates how large-scale brain networks, including the DMN, exhibit time-varying connectivity patterns that align with ongoing affective states (Lettieri, G., Handjaras, G., Setti, F., Cappello, E. M., Bruno, V., Diano, M., ... & Cecchetti, L. (2022).”

Answer: Thank you for the suggestions. As suggested, we added this study in the text and used it to enhance the discussion on this topic.

Reviewer #1: “Default and control network connectivity dynamics track the stream of affect at multiple timescales. Social cognitive and affective neuroscience, 17(5), 461-469.). Integrating this perspective would provide a more nuanced discussion of how DMN interactions evolve in real-time rather than being constrained to static connectivity descriptions.”

Answer: Thank you for the suggestions. We added more information about this topic to enhance the discussion on this topic.

Reviewer #1: “The manuscript could be strengthened by incorporating more discussion on the presence of DMN-like activity in non-human species. Comparative neuroscience studies have observed resting-state networks in animals, suggesting evolutionary continuity. A discussion of such studies would provide valuable context for the claims made about DMN function and development.”

Answer: Thank you for the suggestions. We added more information about this topic to enhance the discussion on this topic.

Reviewer #1: “Overall, this manuscript is a valuable contribution to the literature on DMN function and its relevance to mental health. However, integrating critiques of methodological limitations, cross-species comparisons, and multimodal evidence would strengthen the discussion. Addressing these areas would provide a more balanced and comprehensive evaluation of the DMN’s role in cognition and psychopathology.”

Answer: Thank you for the complement for our manuscript. We followed all the suggestion to improve the quality of text. The alterations are highlighted in yellow.

I hope you find the revised version of our manuscript suitable for publication. Thank you in advance for your consideration.

Sincerely,

Prof. Dr. Durvanei Augusto Maria

Butantan Institute

Reviewer 2 Report

Comments and Suggestions for Authors

Paper purpose: Review and synthesize the existing literature on the default mode network.

Comments:

  1. Overall, the paper in its present form would benefit from an additional round of edits from the authors. There are several instances that give the impression that the various sections of the paper may have been written by different authors with minimal integration across sections. Specifically, the default mode network is defined in the same/similar way repeatedly across sections. Acronyms are used inconsistently (e.g., default mode network is often reintroduced repeatedly rather than simply referring to it as “DMN” after the first introduction). There is also inconsistent capitalization (e.g., ‘default mode network’ vs “Default Mode Network”). Another example would be the multiple mentions of childhood stress that could be integrated into a single section. Streamlining this work and minimizing the number of instances of repeated information, would greatly improve readability.
    1. An additional reading of the paper may also be beneficial as there are some claims within the text not well supported by references. Please consider where any strong claims may be better supported with an appropriate reference.

  1. It would be more cohesive of the contents of Tables 1, 2, and Figure 1were describe in narrative form. This would be consistent with the rest of the paper and allow for greater integration and richer discussion of the theories and how technological advancements/ research informed newer theories.

  1. In line with comment 2, the discussion of the relevant brain regions is incredibly short within lines 84-92. Integrating this section with the preceding sections would be beneficial, particularly given the depth of coverage for the secondary brain regions.

  1. Lines 117-120: Do you think that evaluating connectivity/activity in the DMN is a practical screening tool for these conditions?

  1. Figure 2 could use some sources overall
    1. The first line describing the red region, which Brodmann area are you referring to?
  2. Section 5 (beginning on line 260) makes it sound as though there is a causal role between childhood adversity and the DMN. There are additional factors (social, demographic) that could influence the rates of childhood adversity and neurological functioning that could be added into this section to enrich the discussion.
  3. Line 373 mentions unique patterns in children and adolescents. It could be helpful to report on these differences and how they relate to brain development.

Author Response

March 3rd, 2025.

Editors-in-Chief

Biology

Dear Editor,

Please find enclosed the revised version of our manuscript entitled “The Journey of the Default Mode Network: Development, Function and Impact on Mental Health” submitted to Biology.

We addressed all of the reviewer’s comments, which were extremely relevant for improving our manuscript. A point-by-point answer is below addressed for each reviewer in separated letters. The changes asked by Reviewer 2 are highlighted in GREEN in the manuscript.

Reviewer #2: “Specifically, the default mode network is defined in the same/similar way repeatedly across sections.”

Answer: Thank you for the suggestions and the complements. We performed a full revision of the text and performed the required changes to avoid repetition of concepts.

Reviewer #2: “Acronyms are used inconsistently (e.g., default mode network is often reintroduced repeatedly rather than simply referring to it as “DMN” after the first introduction). There is also inconsistent capitalization (e.g., ‘default mode network’ vs “Default Mode Network”).”

Answer: We performed a revision and altered the ‘default mode network’ for DMN to avoid repetition.

Reviewer #2: “Another example would be the multiple mentions of childhood stress that could be integrated into a single section.”

Answer: Thank you for the suggestion. We amended section 5 and 10 transformed into a single section, reorganizing the manuscript.

Reviewer #2: “Please consider where any strong claims may be better supported with an appropriate reference.”

Answer: We added more recent references to support our discussion and explored ideas.

Reviewer #2: “It would be more cohesive of the contents of Tables 1, 2, and Figure 1were describe in narrative form.”

Answer: Thank you for the suggestions. Table 01 turned in narrative form, but able 2 and Figure 1 are still in the original form. We believe that it helps the reader with fluidity in reading, draws attention to the main highlighted points.

Reviewer #2: “In line with comment 2, the discussion of the relevant brain regions is incredibly short within lines 84-92. Integrating this section with the preceding sections would be beneficial, particularly given the depth of coverage for the secondary brain regions.”

Answer: Thank you for the suggestions. The discussion was extended and related to section 3

Reviewer #2: “Lines 117-120: Do you think that evaluating connectivity/activity in the DMN is a practical screening tool for these conditions?”

Answer: We believe that it could be a complementary tool that can be used to identify possible early markers of neurological disorders, such as autism spectrum disorder (ASD) and attention deficit hyperactivity disorder (ADHD).

Reviewer #2: “Figure 2 could use some sources overall”

Answer: We added more in the text.

Reviewer #2: “The first line describing the red region, which Brodmann area are you referring to?”

Answer: Brodmann areas 9 and 10, which are involved in higher cognitive processes such as self-awareness, social decision-making and moral judgment.

Reviewer #2: “Section 5 (beginning on line 260) makes it sound as though there is a causal role between childhood adversity and the DMN. There are additional factors (social, demographic) that could influence the rates of childhood adversity and neurological functioning that could be added into this section to enrich the discussion.”

Answer: Thank you for the suggestions. We added more information on this matter in the text to improved understanding.

Reviewer #2: “Line 373 mentions unique patterns in children and adolescents. It could be helpful to report on these differences and how they relate to brain development.”

Answer: Thank you for the suggestions. We added more information on this matter in the text to improved understanding.

I hope you find the revised version of our manuscript suitable for publication. Thank you in advance for your consideration.

Sincerely,

Prof. Dr. Durvanei Augusto Maria

Butantan Institute

Reviewer 3 Report

Comments and Suggestions for Authors

In the manuscript, the authors reviewed the brief history and the current status of the research on the default mode network (DMN). The review is comprehensive, which will contribute to the literature. Several issues should be addressed:

  1. It would be helpful to add a table to summarize the studies that were reviewed in the article.
  2. Section 2 and 3 are largely overlapping, please consider to merge the sections.
  3. In section 4, the limbic system is not a brain network, please clarify this statement.
  4. Please consider merge section 5 and 10, given both sections focus on neurodevelopment.
  5. It would be helpful to suggest future directions for the research on DMN.
  6. Minor: in-text citations should use the same format, e.g., "(Andrews-Hanna et al., 2010; Diez et al., 2015; Hausman et al., 2020; Wang et al., 2020)" in the second paragraph on p.2.

Author Response

March 3rd, 2025.

Editors-in-Chief

Biology

Dear Editor,

Please find enclosed the revised version of our manuscript entitled “The Journey of the Default Mode Network: Development, Function and Impact on Mental Health” submitted to Biology.

We addressed all of the reviewer’s comments, which were extremely relevant for improving our manuscript. A point-by-point answer is below addressed for each reviewer in separated letters. The changes asked by Reviewer 3 are highlighted in BLUE in the manuscript.

Reviewer #3: “It would be helpful to add a table to summarize the studies that were reviewed in the article.”

Answer: After considering your proposal, we decided not to include a table summarizing the reviewed studies. We understand that the suggestion is pertinent, however, we believe that, since this is a literature review, the current format of the article already provides a sufficient and detailed analysis of the selected studies. Therefore, we do not see the need to add the table at this time.

Reviewer #3: “Section 2 and 3 are largely overlapping, please consider to merge the sections.”

Answer: The sections were merged to improve the manuscript organization as suggested. Thank you.

Reviewer #3: “In section 4, the limbic system is not a brain network, please clarify this statement.”

Answer: We apologize for the mistake. It was amended in the text.

Reviewer #3: “Please consider merge section 5 and 10, given both sections focus on neurodevelopment.”

Answer: The sections were merged to improve the manuscript organization as suggested. Thank you.

Reviewer #3: “It would be helpful to suggest future directions for the research on DMN.”

Answer: Thank you for the suggestion. It was added a paragraph discussing this matter in the end of the manuscript.

Reviewer #3: “Minor: in-text citations should use the same format, e.g., "(Andrews-Hanna et al., 2010; Diez et al., 2015; Hausman et al., 2020; Wang et al., 2020)" in the second paragraph on p.2.”

Answer: Corrected as requested. Thank you for the revision.

I hope you find the revised version of our manuscript suitable for publication. Thank you in advance for your consideration.

Sincerely,

Prof. Dr. Durvanei Augusto Maria

Butantan Institute

Round 2

Reviewer 2 Report

Comments and Suggestions for Authors

Author Response

March 17th, 2025.

Editors-in-Chief

Biology

Dear Editor,

Please find enclosed the revised version of our manuscript entitled “The Journey of the Default Mode Network: Development, Function and Impact on Mental Health” submitted to Biology.

We addressed all of the reviewer’s comments, which were extremely relevant for improving our manuscript. A point-by-point answer is below addressed for each reviewer in separated letters. The changes asked by Reviewer 2 are highlighted in YELLOW in the manuscript.

Reviewer #2: “Consistent use of acronyms. It is clear that the authors made several adjustments, which are appreciated. However, there are still several areas where “Default Mode Network (DMN)” is repeated in the text.

  1. Other examples include:
    1. Generalized anxiety disorder (GAD)
    2. Functional magnetic resonance imaging (fMRl)
  • Cognitive behavioral therapy (CBT)”

Answer: Thank you for the corrections. We performed the requested alterations for acronyms consistency.

Reviewer #2: “Repeated text. There are several areas where text is repeated word for word or previously introduced concepts are redefined. Please read through again to ensure the manuscript is not overly repetitive. Specific examples include:

  1. Lines 232- 235 and lines 229—231
  2. Several elements within the “Neuroanatomical Basis of the Default Mode Network” section
  3. Lines 605— 610
  4. Section 6.7 should include your discussion of childhood stress earlier, the two separate sections are not needed and could be integrated
  5. Sections 4.1 and 4.2 can be integrated into the earlier paragraphs of section 4
  6. Sections 6.2 and 6.8 can be integrated
  7. Lines 1078 — 1080”

Answer: Thank you for the pointed corrections. We performed the requested alterations.

Reviewer #2: “I like the background information in section 9. I would consider moving it up earlier in the manuscript to provide valuable background information for the reader.”

Answer: Thank you for the pointed corrections. We performed the requested alterations

Reviewer #2: “Line 295 ends abruptly”

Answer: Thank you for the pointed corrections. We performed the requested alterations.

Reviewer #2: Please add that you are referring to Brodmann Areas 9 and 1 0 within the mPFC section of Figure 2.”

Answer: Thank you for the pointed corrections. We performed the requested alterations.

Reviewer #2: “I don't think lines 525-545 are needed, at least not in this section. It feels a bit out of place.”

Answer: Thank you for the pointed corrections. We performed the requested alterations.

  1. Reviewer #2: Please consider the use of more affirming language. The APA has some helpful guidelines on best practices to refer to individuals with various diagnoses. Some suggestions include:
    1. Line 894: “Medical and Psychological Diagnoses” instead of “Mental Disorders”
    2. Line 954: “individuals with schizophrenia” rather than “schizophrenic patients”
    3. Section 7.4: “individuals with a diagnosis of PTSD” rather than “PTSD patients”
    4. Line 1069: “psychiatric diagnoses” instead of “mental illnesses”
    5. Line 1096: “psychiatric diagnoses” instead of “mental disorders”

Answer: Thank you for the pointed corrections. Affirming language is quite important regarding this topic, we performed the requested alterations.

I hope you find the revised version of our manuscript suitable for publication. Thank you in advance for your consideration.

Sincerely,

Prof. Dr. Durvanei Augusto Maria

Butantan Institute

Round 3

Reviewer 2 Report

Comments and Suggestions for Authors

Thank you for the edits and updates. I would encourage an additional read through of the manuscript to ensure it is as streamlined and non-redundant as possible. 

Author Response

Thank you very much for all your revisions on the manuscript. We believe that they have significantly enhanced the quality of the text. As requested, we have conducted a thorough review and are confident that it is now ready for publication. We hope this meets the necessary requirements. Once again, thank you for your invaluable assistance.